# Nature and Mental Health in Urban Texas: A NatureScore-Based Study

**DOI:** 10.3390/ijerph21020168

**Published:** 2024-02-01

**Authors:** Omar M. Makram, Alan Pan, Jay E. Maddock, Bita A. Kash

**Affiliations:** 1Center for Health & Nature, Houston Methodist Research Institute, Houston, TX 77030, USA; omarmakram95@gmail.com (O.M.M.);; 2Center for Health Data Science and Analytics, Houston Methodist Research Institute, Houston, TX 77030, USA; apan@houstonmethodist.org; 3Department of Environmental and Occupational Health, School of Public Health, Texas A&M University, 1266 TAMU, College Station, TX 77843, USA

**Keywords:** mental health, stress, depression, NatureScore, nature, urban design

## Abstract

In this cross-sectional study, we examined the impact of access to nature on mental health utilization in urban neighborhoods using Texas outpatient encounters data merged with NatureScore^TM^ (0–100; low to high nature levels) and US census data (household income, education, employment, poverty, and insurance coverage) at the zipcode level. Our sample size included 61 million outpatient encounters across 1169 zipcodes, with 63% women and 30% elderly. A total of 369,344 mental health encounters were identified, with anxiety/stress and depression encounters representing 68.3% and 23.6%, respectively. We found that neighborhoods with a NatureScore of 60+ had lower overall mental health utilization than those below 40 (RR 0.51, 95%CI 0.38–0.69). This relationship persisted for depression, bipolar disorder, and anxiety/stress and in neighborhoods with a NatureScore above 80 (*p* < 0.001). Compared to neighborhoods with a NatureScore below 40, those above 80 had significantly lower depression (aRR 0.68, 95%CI 0.49–0.95) and bipolar (aRR 0.59, 95%CI 0.36–0.99) health encounters after adjusting for demographic and socioeconomic factors. This novel approach, utilizing NatureScore as a proxy for urban greenness, demonstrates the correlation between a higher NatureScore and reduced mental health utilization. Our findings highlight the importance of integrating nature into our healthcare strategies to promote well-being and mental health.

## 1. Introduction

Nature has been an essential component of human life for thousands of years. It has played a critical role in human development and social experience. Interaction with the natural environment, like parks and forests, has significantly impacted both physical and mental health [1]. Health benefits occur both through immersive nature experiences, such as forest bathing [2], and also shorter, less intense exposures, including urban nature [3,4]. Access to nature, especially in urban areas, also promotes increased physical activity [5].

The prevalence of mental health disorders in the United States has been surging over the last few years, affecting more than 22% of the adult population [6]. With a substantial increase in mental health issues, a few studies have found a relationship between the various social determinants of health and mental health outcomes [7,8]. A significant association between urban green space and improved mental health has been found in Australia, Finland, and Florida, USA [9,10,11]. However, defining nature and relying on subjective measures in these studies might limit comprehension of the full-scale impact of nature access and the built environment on human health.

In this study, we aimed to explore the relationship between access to nature in urban neighborhoods across Texas, measured in the form of NatureScore, and mental health visits. To our knowledge, only one peer-reviewed paper has been published using NatureScore as a comprehensive measure of nature exposure by geographic location [12]. Most studies of nature and mental health outcomes have only been able to capture one or two elements of nature at low granularity (e.g., average normalized difference vegetation index—NDVI—or tree canopy) at a time. The complex nature of NatureScore allows for a more comprehensive look at nature exposure and its potential benefits on mental health.

## 2. Materials and Methods

### 2.1. Setting and Study Design 

#### 2.1.1. Texas Outpatient Encounters Data

A retrospective cohort study design was conducted using Texas Hospital Outpatient Public Use Data Files from 2014 to mid-2019 [13]. The data were aggregated at the zipcode level and contained de-identified patient encounters. The dataset included age, gender, race/ethnicity, principal diagnosis, and zipcode.

#### 2.1.2. Rural-Urban Commuting Area (RUCA) Codes 

In this study, we limited our analysis to urban areas in Texas. For this purpose, we used the 2010 Rural–Urban Commuting Area (RUCA) codes provided by the US Department of Agriculture Economic Research Service (USDA-ERS) [14]. RUCA codes not only sort geographical areas based on the population, the commuting patterns also “flow” into these areas. Thus, we used the RUCA codes (1–3) corresponding to metropolitan areas irrespective of the volume of commuting to urban areas [15,16]. The RUCA codes represent the urban–rural definition at the zipcode level. 

#### 2.1.3. U.S. Census Data

The five-year averaged estimates (2016–2020) collected as part of the U.S. Census Bureau’s American Community Survey included various socioeconomic factors at the zipcode level. The collected factors included health insurance coverage, educational attainment, employment status (currently employed or not), median household income, and poverty level [17]. In our study, we defined educational attainment as being at least 25 years old and having at least a bachelor’s degree. Poverty was defined according to the number of household members and the poverty level defined by the Census Bureau for a specific year. With all measures collected at the zipcode level, the data were merged with Texas outpatient data.

### 2.2. Study Population

In our study, we used a cross-sectional study design approach, and the unit of observation was the zipcode. The STROBE checklist for cross-sectional studies was used to conform with the guidelines for reporting results from observational studies [18] (Appendix A). Also, a complete case analysis approach was taken during the regression analysis; thus, zipcodes with available outpatient data on the assigned outcome, US census data, and RUCA information were included in the regression analysis. At the encounter level, we initially included 92,681,810 mental health outpatient encounters across all zipcodes, representing six years of data (2014 to mid-2019). A total of 18,050,949 outpatient encounters were excluded before the data aggregation at the zipcode level, and finally, a sample of 1169 zipcodes (n = 61,391,400 adult outpatient encounters) was included (Figure 1).

### 2.3. Study Variables

#### 2.3.1. Exposure of Interest: NatureScore

NatureScore is a dynamic measure of the amount and quality of natural elements of any point or polygon using a patent-pending system created by NatureQuant [19]. NatureScore is an improvement over other single-measure indicators on greenspace like NDVI and canopy cover by creating a holistic picture of nature in each area. The datasets include a broad array of environmental features, including satellite images of vegetation to land use cover and classifications, parks, tree canopy cover, noise levels, artificial light, air pollution, buildings, roads, and aerial and street view images. The data are weighted and summed to create an overall NatureScore^TM^ value based on machine learning models [20]. The NatureScore values range from zero (poor NatureScore, lacking beneficial natural elements) to 100 (high NatureScore, abundant beneficial natural elements) (Appendix A). For this study, the NatureScore data were calculated via an examination of the elements within the provided zipcode polygons based on 2019 data, and the data were categorized into four groups: Nature Deficient/Nature Light (0–39), Nature Adequate (40–59), Nature Rich (60–79), and Nature Utopia (80–100).

In a nationwide U.S. census tract-based examination, NatureScore was previously validated against NDVI, and a strong correlation (r = 0.87) was found to be present [12].

#### 2.3.2. Outcome of Interest: Mental Health Encounters per 100,000 Population

International Classification of Diseases 9th Revision (ICD-9) and International Classification of Diseases 10th Revision Clinical Modification (ICD-10 CM) codes were used to identify various mental health encounters (depression, bipolar disorders, stress, and anxiety) using the principal diagnosis variable (Appendix A). Post-traumatic stress disorders, acute stress disorders, and adjustment disorders were all defined as stress disorders. Anxiety and panic disorders were defined as anxiety disorders. The rates of mental health encounters were calculated by dividing the number of specific mental health encounters within a certain zipcode by the total population (18+ years) in that zipcode. Finally, rates were standardized per 100,000 population.

#### 2.3.3. Covariates

Other variables, such as demographics (age, gender, and race/ethnicity) and socioeconomic factors (educational attainment, employment status, and poverty level), were included as covariates in the final regression model. In our model, we categorized age into three categories (18–44, 45–64, and 65+ years). Elderly status was defined as being 65+ years of age. The percentage of each variable in each zipcode was used.

### 2.4. Statistical Analyses

Study characteristics were presented using either mean and standard deviation (SD), median and interquartile range, or percentages for normally distributed continuous data, non-normally distributed continuous data, and categorical data, respectively. To test the difference in data distribution across the four categories of NatureScore, one-way analysis of variance (ANOVA) and Kruskal–Wallis tests were conducted for continuous normal and non-normal data, respectively.

A correlation matrix was conducted before regression analysis to explore the relationship between the socioeconomic factors and the NatureScore and to avoid multi-collinearity (Appendix A). In this matrix, we utilized the Pearson correlation coefficient (r), which ranges from −1 to 1, where 1 indicates a perfect positive correlation, and −1 indicates a perfect negative correlation. Univariable and multivariable generalized linear models (GLM) were built to investigate our research question. A Box–Cox distribution and modified Park tests were used to determine the appropriate link function and family to use in the model, respectively. A log-link function and the inverse Gaussian family were used in all the GLM models [20]. To adjust for the individuals’ similarities within a certain zipcode (clustering), robust standard errors were used. In the final model, we adjusted for demographic and socioeconomic factors. Testing for interaction was conducted in the final regression model, and stratification was performed if significant. Results were presented in the exponential form, representing rate ratio (RR) and adjusted rate ratio (aRR). Lastly, Stata/MP 17.0 (StataCorp, College Station, TX, USA) software was used to conduct all the statistical analyses. Results were deemed statistically significant if the two-sided *p*-value was <0.05.

## 3. Results

### 3.1. NatureScore

Our analytical sample included data from 1169 zipcodes in urban Texas, with a median NatureScore of 85.8. About half of our sample had high NatureScores (80+), and about 22% of zipcodes had NatureScore below 40 (Appendix A). 

From the included encounters, a total of 369,344 mental health encounters were identified. These encounters were divided into anxiety/stress (68.3%, n = 252,170), depression (23.6%, n = 87,052), and bipolar (8.1%, n = 30,122) encounters. We found that the rate of mental health encounters was 2532 per 100,000 population at the zipcode level. The highest rate of encounters was found in anxiety/stress encounters (1787 per 100,000 population), followed by depression (548 per 100,000 population) and bipolar encounters (196 per 100,000 population) (Table 1).

### 3.2. Demographic and Socioeconomic Factors

Of the total mental health encounters, 63% were women, 30% were elderly, 54% were non-Hispanic whites, and 15% were Hispanics. Using US census data, we found that at the zipcode level, 27% of the total population had a bachelor’s degree, 58% were employed, 14% lived under poverty, and 17% lacked health insurance coverage. Significant differences were found for demographics and socioeconomic factors between the four groups of NatureScore. The percentage of elderly, Whites, Hispanics, and employed individuals were higher in areas with a higher NatureScore. On the other hand, the zipcodes with a higher NatureScore had lower percentages of Blacks, poverty, and lack of insurance (Table 1).

There were strong correlations between median household income and educational attainment (r = 0.72), lack of insurance (r = −0.61), and poverty (r = −0.69). Also, the analysis showed strong correlations between lack of insurance and poverty (r = 0.60) and lack of insurance and educational attainment (r = 0.59). A modest negative correlation was found between NatureScore and poverty (r = −0.31). Based on these findings, we excluded median household income and lack of insurance from the final multivariable regression model. In the final model, a moderate correlation was found between educational attainment and poverty (r = −0.44); educational attainment and employment (r = 0.43); and poverty and employment (r = −0.42) (Appendix A). 

The univariable analysis demonstrated that the neighborhoods with higher percentages of women had significantly lower rates of depression (RR 0.013, 95%CI 0.002–0.074) and bipolar (RR 0.005, 95%CI 0.000–0.076) outpatient encounters. Also, neighborhoods with higher proportions of elderly showed lower rates of both depression (RR 0.168, 95%CI 0.075–0.377) and bipolar disorder (RR 0.224, 95%CI 0.055–0.919). Zipcodes comprising higher proportions of White (RR 0.5, 95%CI 0.345–0.724) and Asian (RR 0.001, 95%CI 0.000–0.003) races were observed to have significantly lower rates of any mental health outpatient encounters. A similar relationship was found in specific mental health outpatient visits (*p* < 0.05). Lastly, higher employment was associated with lower rates of any mental health encounters (RR 0.99, 95%CI 0.983–0.998), as well as specifically anxiety/stress (RR 0.987, 95%CI 0.979–0.995) outpatient encounters (Table 2).

Contrastingly, zipcodes with a higher proportion of younger (18–44 years) individuals were significantly associated with a higher number of depression encounters (RR 4.434, 95%CI 1.507–13.049). Similar findings were found between the Black population and depression and bipolar visits and the Hispanic population and both anxiety/stress disorder encounters and any mental health encounters (*p* < 0.05). Lastly, a higher poverty ratio was associated with slightly higher yet statistically significant overall mental health encounters (RR 1.013, 95%CI 1.005–1.021) and the various mental health-specific encounters: depression (RR 1.013, 95%CI 1.004–1.023), bipolar disorder (RR 1.017, 95%CI 1.002–1.033), and anxiety/stress disorders (RR 1.012, 95%CI 1.004–1.020) (Table 2).

In the multivariable regression analysis, zipcodes with higher female (aRR 0.107, 95%CI 0.025–0.468) and Asian representations (aRR 0.004, 95%CI 0.000–0.026) were associated with lower rates of any mental health encounters. The same association was found between both variables and the specific mental health outcomes (depression, bipolar, and anxiety/stress disorders) (*p* < 0.05) (Table 2). Other factors, such as a higher Hispanic population, showed a significant relationship with lower depression encounters (aRR 0.584, 95%CI 0.387–0.881), and higher educational attainment was associated with lower rates of any mental health encounters (aRR 0.994, 95%CI 0.989–0.999), bipolar disorders (aRR 0.989, 95%CI 0.983–0.994), and anxiety/stress disorders (aRR 0.993, 95%CI 0.988–0.998). However, we found that elderly status was associated with higher bipolar encounters (aRR 8.513, 95%CI 1.688–42.948), and the Hispanic population was associated with higher overall mental health encounters (aRR 2.216, 95%CI 1.516–3.241) and anxiety/stress disorders (aRR 3.502, 95%CI 2.315–5.296). Also, the positive association between the Black populations and mental health encounters remained significant after adjusting for all other factors (*p* < 0.01). Lastly, the observed association between higher poverty ratios and increased mental health encounters was no longer significant after adjusting for demographic and socioeconomic factors, except for bipolar encounters (aRR 1.019, 95%CI 1.007–1.032) (Table 2).

### 3.3. Mental Health and NatureScore

We demonstrated a significant difference in the rates of overall and specific mental health encounters across the categories of neighborhood NatureScores. Neighborhoods with NatureScores over 60 showed about 50% lower rates of metal health encounters than those below 60 (Table 1). Additionally, we observed a decreasing trend in the various mental health encounters as the NatureScore of a neighborhood increased (Figure 2).

In the univariable regression models, both Nature Rich (RR 0.510, 95%CI 0.378–0.689) and Nature Utopia (RR 0.569, 95%CI 0.419–0.771) neighborhoods demonstrated significantly lower rates of mental health encounters, compared to neighborhoods with the lowest NatureScore category. The same finding was observed in specific mental health outcomes (depression, bipolar, and anxiety/stress) in both NatureScore categories of Nature Rich and Utopia (*p* < 0.01). The regression analysis also showed that compared to the lowest NatureScore neighborhoods, neighborhoods with a NatureScore just above 40 (Nature Adequate) had at least 51% and 63% lower likelihoods of depression and bipolar encounters, respectively (Table 2).

In the multivariable regression analysis, we found that neighborhoods with the highest NatureScore (Nature Utopia) had lower rates of depression (aRR 0.683, 95%CI 0.490–0.950) and bipolar (aRR 0.594, 95%CI 0.355–0.994) outpatient encounters when compared to neighborhoods with a NatureScore below 40. When comparing neighborhoods with a NatureScore of 60–79 with neighborhoods with a NatureScore below 40, we found that areas with higher scores had less mental health outpatient encounters but with no statistical significance (Table 2) (Figure 3). 

## 4. Discussion

This is the first study that investigated the relationship between access to nature in urban neighborhoods, in the form of NatureScore, and mental health outcomes. In this study, we demonstrated a significant association between exposure to nature and mental health visits. Based on the multivariate analysis, this study also confirms the wide disparities in mental health utilization by race/ethnicity. We found that a neighborhood with a NatureScore above 40 (Nature Adequate) has 51% and 63% lower likelihoods of depression and bipolar encounters than those below 40, respectively. Nevertheless, after controlling for all covariates and comparing neighborhoods with a NatureScore above 80 versus those below 40, we found 32% and 41% lower likelihoods of depression and bipolar encounters, respectively. This translates to a potential meaningful NatureScore threshold of 40 to be considered when planning and improving urban design. To illustrate this threshold, we included satellite images of neighborhoods in different NatureScore categories (Appendix A). 

Consistent with our findings, a smaller study in Florida found that higher levels of greenness using NDVI were associated with lower odds of depression in individuals above the age of 65 [11]. Another study in Finland examined the cumulative effects of residential greenness and the odds of depression reported in 5-year and 14-year follow-up periods. In this study, they found a significant association between NDVI-calculated greenness score and lower odds of doctor-diagnosed depression, even after adjusting for age, gender, marital status, education, employment, BMI, and chronic diseases [10].

It is also important to observe how the quality of green space and individuals’ perceptions of their neighborhood might affect their mental health outcomes. In a study conducted in Australia on 3897 postpartum women and following them up for 15 years, they investigated the relationship between the participants’ perceptions of green space quality and the incidence of serious mental illness. They found significantly lower rates of psychological distress and serious mental illness in women who agreed or strongly agreed that local parks were of good quality [21]. Another study in Australia examined the relationship between the type of green space in cities and psychological distress, stratified by the type of housing. They found that tree canopy was associated with lower odds of psychological distress in both apartment- and house-dwellers. On the other hand, open grass spaces were associated with higher odds of psychological distress among the participants of both groups [9].

The mechanism explaining the relationship between greenness and mental health is quite complex. Previous studies have demonstrated the impact of nature walks on improving the mood and attention and reducing stress [22,23]. Also, the amount of time required to observe a significant impact is quite variable. One study has demonstrated that spending >120 min/week in nature was associated with reporting better health and well-being [24]. Others have shown that significant stress relief and a reduction in salivary cortisol occurred when spending between 20–30 min per exposure in urban nature [25]. Nevertheless, green spaces and urban nature encourage physical activity, which in turn improves overall health [26,27]. We believe further research is required to explain that complex relationship and to assess the cost-effectiveness of the various built environmental interventions to increase the greenness.

One of the limitations of the NatureScore calculated at the zipcode level in this study is that it is less accurate than using a specific NatureScore for each specific address. Further studies need to investigate the use of NatureScore as an exposure using smaller footprints of examination, e.g., individual addresses with a 250–500 m radius around the home, smaller grids for measurements (50–100 m grids), or measurements at the census block level. Another limitation comes from the lack of follow-up due to data confidentiality, thus limiting our comprehension about the cause–effect relationship and establishing causality. In our study, we defined urban–rural areas using the RUCA codes, yet other systems of identification exist but at different scales of observations (e.g., census tracts and blocks). With different systems of identification, an overlap of definitions was inevitable [28,29]. 

While we employed the principal diagnosis codes to enhance accuracy, some limitations may still exist. Specifically, the dataset included patients who underwent any radiological assessments or surgical procedures during the outpatient encounter, potentially introducing bias by excluding patients without these procedures. Also, our data were limited to the period from 2014 to 2019 in Texas; thus, the relationship between nature access and mental health might be impacted by COVID-19 starting in 2020 or by data from other US states with different demographics or behaviors. It is essential to note that the NatureScore data do not consider certain variables, such as physical activity, safety, and human interaction with nature, which could potentially impact the relationship between access to nature and human health. Lastly, despite accounting for most of the available demographic and socioeconomic factors, residual confounding might still exist. 

## 5. Conclusions

In conclusion, this is the first study to utilize NatureScore as a proxy for urban greenness and study its correlation with mental health. Our results indicate that a higher NatureScore is associated with better mental health outcomes. Increasing green space in cities may provide another avenue to address the well-documented shortage of mental health professionals [30]. This study establishes the foundations for future research into the use of NatureScore as an all-encompassing measure of nature exposure and its impact on various health outcomes. Overall, our findings highlight the importance of incorporating nature into our built environment and healthcare strategies to promote well-being and mental health.

## Figures and Tables

**Figure 1 ijerph-21-00168-f001:**
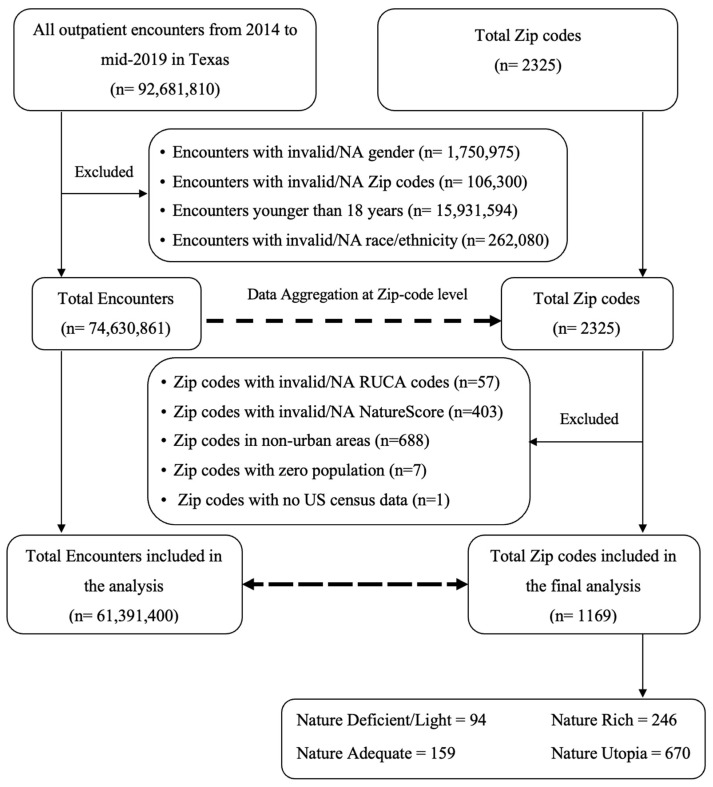
Flowchart depicting the inclusion process for the population in the final analysis.

**Figure 2 ijerph-21-00168-f002:**
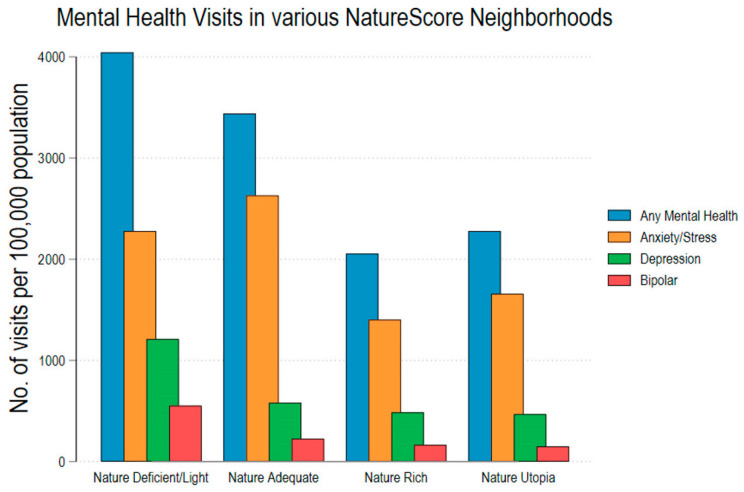
Bar chart illustrating differences in mental health encounters across different NatureScore neighborhoods.

**Figure 3 ijerph-21-00168-f003:**
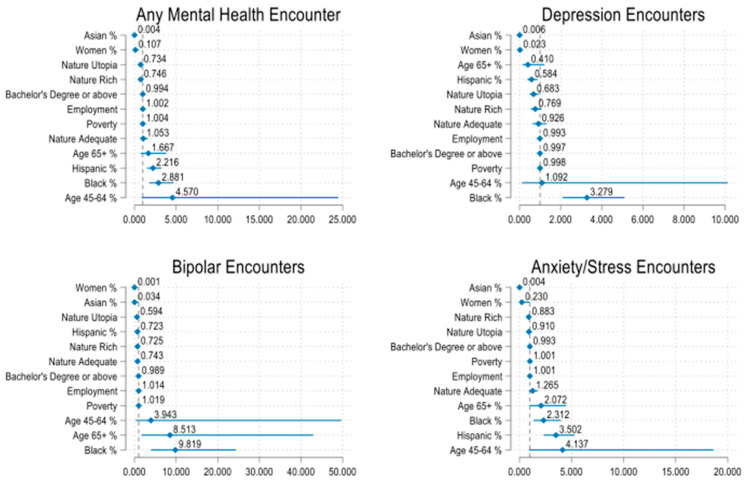
Forest plot of multivariable regression results for different mental health encounters, sorted by factors influencing mental health outcomes from lowest to highest.

**Table 1 ijerph-21-00168-t001:** Distribution of demographics, socioeconomic factors, and mental health encounters across NatureScore categories.

	Summary Statistics, Mean (SD)					
	Total	Nature Deficient/Nature Light	Nature Adequate	Nature Rich	Nature Utopia	*p*-Value ^#^
No. of Zipcodes	1169	94 (8%)	159 (14%)	246 (21%)	670 (57%)	
Total outpatient encounters	61,391,398	3,826,697 (6.2%)	10,485,831 (17.1%)	18,408,516 (30%)	28,670,354 (46.7%)	
NatureScore ^+^	85.8 (64.8–94.7)	30.35 (19.5–36.8)	50.3 (46.7–57.0)	71.5 (66.6–75.9)	93.4 (89.1–97.4)	<0.001 ^+^
Demographics						
Total population (18+) from Census data, No.	18,677,889	1,189,009	3,289,630	5,870,487	8,328,763	
Population (18+) per Zipcode, Mean (SD)	15,978 (15,566)	12,649 (14,181)	20,689 (15,529)	23,864 (15,494)	12,431 (14,419)	<0.001
Women, %	63.07 (4.38)	60.72 (8.02)	64.25 (4.25)	64.62 (3.43)	62.55 (3.70)	<0.001
Age 18–44, %	34.21 (10.35)	39.96 (13.41)	38.06 (11.28)	36.65 (9.68)	31.60 (8.96)	<0.001
Age 45–64, %	35.82 (4.83)	34.77 (7.22)	35.16 (5.70)	36.23 (4.61)	35.97 (4.20)	0.02
Age 65+, %	29.97 (9.25)	25.27 (10.54)	26.78 (9.15)	27.12 (8.35)	32.43 (8.61)	<0.001
White, %	54.09 (24.98)	36.17 (19.85)	39.51 (21.47)	45.46 (23.22)	63.23 (22.83)	<0.001
Black, %	12.15 (15.96)	11.37 (12.85)	12.61 (14.58)	16.70 (18.24)	10.48 (15.46)	<0.001
Asian, %	1.63 (2.57)	1.34 (1.79)	2.58 (3.92)	2.52 (2.89)	1.12 (1.91)	<0.001
Hispanic, %	15.21 (17.79)	31.12 (23.64)	22.48 (22.02)	14.72 (16.35)	11.43 (14.15)	<0.001
Socioeconomic Factors						
Bachelor’s degree or above, %	26.79 (18.07)	27.94 (21.28)	27.65 (20.47)	32.14 (19.46)	24.46 (15.91)	<0.001
Employment, %	58.36 (11.41)	55.74 (15.85)	59.87 (12.16)	62.14 (9.17)	56.98 (10.86)	<0.001
Poverty, %	14.43 (10.64)	20.89 (12.74)	18.65 (12.87)	14.68 (9.40)	12.45 (9.46)	<0.001
Median Household Income, $	66,268 (27,112)	52,289 (22,075)	57,140 (23,427)	69,468 (32,471)	69,237 (25,261)	<0.001
Lack of insurance, %	17.03 (9.25)	19.61 (9.88)	19.79 (9.76)	17.89 (9.46)	15.71 (8.70)	<0.001
Mental Health Encounters, per 100,000 population *						
Any Mental Illness	2532.17 (4532.91)	4044.86 (5859.35)	3439.97 (9807.87)	2056.48 (1141.47)	2279.16 (2690.77)	<0.001
Depression	548.35 (946.24)	1212.44 (2622.31)	582.41 (1031.99)	487.42 (334.89)	469.48 (494.84)	<0.001
Bipolar	196.54 (579.02)	553.25 (1647.49)	226.69 (588.05)	165.74 (128.29)	150.64 (315.85)	<0.001
Anxiety/Stress	1787.28 (3858.40)	2279.16 (2001.52)	2630.87 (9128.61)	1403.33 (868.01)	1659.04 (2281.28)	0.006

No: number; SD: standard deviation; %: percentage; ^#^ one-way ANOVA test conducted between the four categories of NatureScores; ^+^ results presented in terms of median and interquartile range (IQR) and Kruskal–Wallis used to compare the four categories of NatureScores; * number of encounters per 100,000 population at the zipcode level.

**Table 2 ijerph-21-00168-t002:** Multivariable regression results for different mental health encounters.

**Univariable Regression Analysis**				
	**Any Mental Health**	**Depression**	**Bipolar**	**Anxiety/Stress**
	**RR (95%CI)**	**RR (95%CI)**	**RR (95%CI)**	**RR (95%CI)**
No. of Zipcodes	1161 ^$^	1118 ^+^	1017 ^#^	1159 ^β^
NatureScore Categories				
Nature Deficient/Nature Light	Reference	Reference	Reference	Reference
Nature Adequate	0.856 (0.504–1.453)	0.485 ** (0.291–0.811)	0.368 ** (0.180–0.751)	1.162 (0.660–2.046)
Nature Rich	0.510 ** (0.378–0.689)	0.405 ** (0.260–0.630)	0.265 ** (0.145–0.482)	0.618 ** (0.510–0.750)
Nature Utopia	0.569 ** (0.419–0.771)	0.401 ** (0.258–0.623)	0.270 ** (0.146–0.498)	0.737 ** (0.600–0.904)
Demographics				
Women %	0.220 (0.017–2.896)	0.013 ** (0.002–0.074)	0.005 ** (0.000–0.076)	1.030 (0.059–17.959)
Age 18–44 %	0.838 (0.219–3.203)	4.434 ** (1.507–13.049)	3.588 (0.654–19.681)	0.335 (0.091–1.239)
Age 45–64 %	4.977 (0.269–91.987)	4.495 (0.213–94.769)	10.103 (0.055–1844.009)	3.487 (0.124–98.133)
Age 65+ %	0.891 (0.256–3.102)	0.168 ** (0.075–0.377)	0.224 * (0.055–0.919)	2.879 (0.704–11.765)
White, %	0.500 ** (0.345–0.724)	0.577 ** (0.415–0.802)	0.528 * (0.290–0.959)	0.497 ** (0.325–0.761)
Black %	1.655 (0.940–2.915)	5.390 ** (2.667–10.895)	7.225 ** (2.722–19.178)	0.963 (0.545–1.703)
Asian %	0.001 ** (0.000–0.003)	0.004 ** (0.001–0.015)	0.001 ** (0.000–0.004)	0.000 ** (0.000–0.002)
Hispanic %	3.091 ** (1.722–5.548)	0.720 (0.455–1.141)	0.901 (0.370–2.192)	4.571 ** (2.576–8.110)
Socioeconomic Factors				
Employment %	0.990 * (0.983–0.998)	0.996 (0.987–1.004)	0.997 (0.988–1.007)	0.987 ** (0.979–0.995)
Bachelor’s degree or above %	0.995 (0.987–1.003)	0.998 (0.991–1.004)	0.998 (0.990–1.005)	0.994 (0.985–1.003)
Poverty %	1.013 ** (1.005–1.021)	1.013 ** (1.004–1.023)	1.017 * (1.002–1.033)	1.012 ** (1.004–1.020)

**Multivariable Regression Analysis**				
	**Any Mental health**	**Depression**	**Bipolar**	**Anxiety/Stress**
	**aRR (95%CI)**	**aRR (95%CI)**	**aRR (95%CI)**	**aRR (95%CI)**
No. of Zipcodes	1159	1116	1016	1157
NatureScore Categories				
Nature Deficient/Nature Light	Reference	Reference	Reference	Reference
Nature Adequate	1.053 (0.689–1.608)	0.926 (0.643–1.332)	0.743 (0.439–1.259)	1.265 (0.901–1.778)
Nature Rich	0.746 (0.512–1.087)	0.769 (0.546–1.082)	0.725 (0.439–1.198)	0.883 (0.690–1.131)
Nature Utopia	0.734 (0.507–1.064)	0.683 * (0.490–0.950)	0.594 * (0.355–0.994)	0.910 (0.712–1.163)
Demographics				
Women %	0.107 ** (0.025–0.468)	0.023 ** (0.004–0.125)	0.001 ** (0.000–0.016)	0.230 * (0.054–0.977)
Age 45–64 %	4.570 (0.852–24.501)	1.092 (0.118–10.124)	3.943 (0.313–49.670)	4.137 (0.918–18.636)
Age 65+ %	1.667 (0.729–3.816)	0.410 (0.139–1.206)	8.513 ** (1.688–42.948)	2.072 (0.964–4.456)
Black %	2.881 ** (1.765–4.704)	3.279 ** (2.105–5.109)	9.819 ** (3.964–24.323)	2.312 ** (1.338–3.995)
Asian %	0.004 ** (0.000–0.026)	0.006 ** (0.001–0.046)	0.034 * (0.001–0.952)	0.004 ** (0.000–0.030)
Hispanic %	2.216 ** (1.516–3.241)	0.584 * (0.387–0.881)	0.723 (0.435–1.201)	3.502 ** (2.315–5.296)
Socioeconomic Factors				
Employment %	1.002 (0.993–1.012)	0.993 (0.980–1.007)	1.014 (0.999 - 1.030)	1.001 (0.993–1.009)
Bachelor’s degree or above %	0.994 * (0.989–0.999)	0.997 (0.991–1.003)	0.989 ** (0.983 - 0.994)	0.993 ** (0.988–0.998)
Poverty %	1.004 (0.995–1.013)	0.998 (0.986–1.010)	1.019 ** (1.007 - 1.032)	1.001 (0.993–1.009)

aRR: adjusted rate ratio; CI: confidence interval; RR: rate ratio; %: percentage. ^$^ The sample size was 1161 for all variables except the poverty ratio (n = 1159), as two zipcodes had zero outcomes; ^+^ the sample size was 1118 for all variables except the poverty ratio (n = 1116), as two zipcodes had zero outcomes; ^#^ the sample size was 1017 for all variables except the poverty ratio (n = 1016), as one zipcode had zero outcomes; ^β^ the sample size was 1159 for all variables except the poverty ratio (n = 1157), as two zipcodes had zero outcomes; * *p* < 0.05, ** *p* < 0.01.

## Data Availability

The datasets analyzed during the current study are publicly available in the Texas Outpatient Public Use Data File (PUDF), https://www.dshs.texas.gov/texas-health-care-information-collection/health-data-researcher-information/texas-outpatient-public-use (accessed on 1 January 2023). Other datasets used for NatureScore can be accessed/purchased through NatureQuant, https://www.naturequant.com/naturescore/ and US Census Bureau, https://data.census.gov/ (accessed on 1 January 2023).

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
