# Peer review of "Nature and Mental Health in Urban Texas: A NatureScore-Based Study"

_ijerph, 2024, doi:10.3390/ijerph21020168_

Round 1

Reviewer 1 Report

Comments and Suggestions for Authors
  1. Need to define outpatient encounters. Does that mean, person went to a hospital as an outpatient?  Does it include both private and public hospitals? Does it also include private mental health practitioners not associated with hospitals? This is important, because relative likelihood of each of these options depends on ethnic, demographic and socioeconomic factors. Readers need to know more about Texas mental health system. IJERPH is an international journal. 

  1. Lots of correlations are quoted as “r”.  All of these should instead be expressed as r-squared, since that is % of variance explained. 

  1. The results and analyses need to be presented differently. The base analysis is outpatient encounters (OE) as a function of socio, demo, ethno factors, ignoring nature. Then the point of this study is, what marginal improvement in prediction of OE is achieved by also including nature score as an additional predictor. And, what marginal improvement is gained from using nature score, rather than more transparent measures of urban greenness.  Only if both of these are significant, is nature score any use.  

Author Response

  1. Need to define outpatient encounters. Does that mean, person went to a hospital as an outpatient? Does it include both private and public hospitals? Does it also include private mental health practitioners not associated with hospitals? This is important, because relative likelihood of each of these options depends on ethnic, demographic and socioeconomic factors. Readers need to know more about Texas mental health system. IJERPH is an international journal. 

Thank you for giving us the opportunity to further explain this point. Exactly, an outpatient encounter means a person went to the hospital as an outpatient and at least received any radiological assessment. However, we only used individuals with the desired outcomes coded as the principal diagnosis to avoid bias. We understand this could be a limitation and added a note about it in the limitations section. "An event or visit for a patient who received services for one or more procedures that included an invasive surgical procedures or imaging/radiological procedures in a hospital or ambulatory surgery center." https://healthdata.dshs.texas.gov/dashboard/hospitals/outpatient-use.

Yes, this includes all hospitals in Texas, including both public and private hospitals, but not private practices not associated with hospitals. However, some exemptions are present as described by the Data source "Exempt hospitals include those located in a county with a population less than 35,000, or those located in a county with a population more than 35,000 and with fewer than 100 licensed hospital beds and not located in an area that is delineated as an urbanized area by the United States Bureau of the Census (Section 108.0025). Exempt hospitals also include hospitals that do not seek insurance payment or government reimbursement (Section 108.009). " 

Data dictionary available through: https://www.dshs.texas.gov/texas-health-care-information-collection/health-data-researcher-information/texas-outpatient-public-use

  1. Lots of correlations are quoted as “r”. All of these should instead be expressed as r-squared, since that is % of variance explained. 

Sorry if this was not clear. but the correlation results presented here are Pearson correlation coefficients (simple correlation) presented as (r), NOT R-squared representing the proportion of variance in the dependent variable that is explained by the independent variable(s) in the model or the goodness of fit in the model. We apologize for the unclear methods used here; it is now corrected in the manuscript. 

  1. The results and analyses need to be presented differently. The base analysis is outpatient encounters (OE) as a function of socio, demo, ethno factors, ignoring nature. Then the point of this study is, what marginal improvement in prediction of OE is achieved by also including nature score as an additional predictor. And, what marginal improvement is gained from using nature score, rather than more transparent measures of urban greenness. Only if both of these are significant, is nature score any use.  

We appreciate your thorough review of our manuscript and the insightful comments regarding the presentation of results and analyses.

We acknowledge your recommendation to explore the marginal effects of NatureScore and compare it to more transparent measures of urban greenness. While our current analysis primarily focused on NatureScore and outpatient encounters (OE) in both univariable and multivariable analyses, incorporating adjustments for various factors, including demographics and socioeconomic variables, we recognize the importance of delving further into these aspects.

It is important to mention that the objective of the study was not to compare model performance or strength, but rather to develop explanatory models that characterize the relative significance of baseline factors (with or without nature-related measures) and their association with mental health outpatient utilization. Future iterations of the work that focus on developing predictive models would need to be trained on datasets that have been enriched or further cleaned to address the limitations of administrative data (e.g., missing information). Also, given the constraints of the current study, we may consider the exploration of marginal effects and even conducting mediation analyses as potential avenues for future research. This could provide a valuable extension to our work, allowing for a more in-depth understanding of the relationships between NatureScore, social determinants of health, and outpatient encounters.

Additionally, we appreciate your suggestion to compare different urban greenness measures, and this will certainly be considered in our future investigations. As we mentioned in our supplementary, NatureScore was only compared to NDVI once, but repeating that in different settings like ours would certainly be valuable and will definitely be considered by our team in future studies.

We are grateful for your constructive feedback, and your suggestions will undoubtedly contribute to the ongoing refinement of our work.

Thank you

Reviewer 2 Report

Comments and Suggestions for Authors

Thank you for the opportunity to review your manuscript, which reports a valuable cross-sectional exploration of mental health utilization according to urban neighbourhood access to nature in Texas as measured with NatureScore, and after controlling for demographic variables available through US census data. While the authors identify the delineation by zip code as a limitation, there is sufficient cause for attention in the findings, which further emphasise the important role of nearby nature in sustaining well-being and mental health.   

The manuscript requires some editorial work to correct and improve written expression and I have some specific comments or queries for you to take into consideration. These are detailed in the attachment. 

Comments on the Quality of English Language

No major issues; some corrections required. 

Author Response

Thank you for your thorough review and positive feedback. We appreciate your valuable suggestions, all of which have been incorporated, along with the additional clarifications requested to improve the manuscript's clarity.

We've also addressed the oversight in Figure 3's title, now accurately reflecting the forest plot's content. It presents the results of the complete final model, organized from lowest to highest factors affecting mental health outcomes, considering all NatureScore categories.

Thank you again and we appreciate any further comments or suggestions.